

# Determining the utility of veterinary tissue archives for retrospective DNA analysis

Firas M. Abed[1] and Michael J. Dark[1,2]

[1] Department of Infectious Diseases and Pathology, College of Veterinary Medicine, University of Florida, Gainesville, FL, United States
[2] Emerging Pathogens Institute, University of Florida, Gainesville, FL, United States

## ABSTRACT

Histopathology tissue archives can be an important source of specimens for retrospective studies, as these include samples covering a large number of diseases. In veterinary medicine, archives also contain samples from a large variety of species and may represent naturally-occurring models of human disease. The formalin-fixed, paraffin-embedded (FFPE) tissues comprising these archives are rich resources for retrospective molecular biology studies and pilot studies for biomarkers, as evidenced by a number of recent publications highlighting FFPE tissues as a resource for analysis of specific diseases. However, DNA extracted from FFPE specimens are modified and fragmented, making utilization challenging. The current study examines the utility of FFPE tissue samples from a veterinary diagnostic laboratory archive in five year intervals from 1977 to 2013, with 2015 as a control year, to determine how standard processing and storage conditions has affected their utility for future studies. There was a significant difference in our ability to obtain large amplicons from samples from 2015 than from the remaining years, as well as an inverse correlation between the age of the samples and product size obtainable. However, usable DNA samples were obtained in at least some of the samples from all years tested, despite variable storage, fixation, and processing conditions. This study will help make veterinary diagnostic laboratory archives more useful in future studies of human and veterinary disease.

# INTRODUCTION

The most commonly used fixative for histopathology is 10% neutral buffered formalin. Formalin preserves tissue and cellular architecture and facilitates subsequent tissue processing. While other fixatives, including TissueTek® Xpress® Molecular Fixative, modified Methacarn, and PAXgene®, have been examined for use in the diagnostic setting (*Craft, Conway & Dark, 2014*), clinical diagnostics are generally dependent on formalin-fixed, paraffin-embedded (FFPE) tissues. Therefore, diagnostic laboratory archives contain large numbers of formalin-fixed samples.

The histopathology archive can be an important source of tissue specimens, as it includes archival tissue specimens from a large number of diseases, and, in veterinary

Corresponding author
Michael J. Dark, darkmich@ufl.edu

medicine, a large variety of species. Depending on the laboratory, it may span a long time frame, making it a valuable resource for retrospective studies and prospective biomarker discovery (*Magdeldin & Yamamoto, 2012*). This is particularly important as it allows linkage of histopathologic findings with molecular information (*Iwamoto et al., 1996*; *Ludyga et al., 2012*), with examination of diseases in various stages of natural disease (*Lewis et al., 2001*).

A number of previous studies have used these for molecular biology research with varying degrees of success, including polymerase chain reaction (PCR), DNA microarrays, and quantitative reverse-transcriptase PCR (qPCR) (*Lewis et al., 2001*). Formalin fixation has a profound effect on nucleic acid quality, with a number of parameters altering the ability to recover usable nucleic acids (*Craft, Conway & Dark, 2014*). In one study using freshly-sampled tissues, formalin-fixed tissues had a mean PCR amplicon size of 300bp, although some samples were able to generate 750bp amplicons (*Craft, Conway & Dark, 2014*). Although the degradation of DNA is severe in some samples, altering the size of the target amplicon can help compensate for degradation (*Iwamoto et al., 1996*; *Lewis et al., 2001*; *Taga et al., 2013*; *Von Ahlfen et al., 2007*). However, the quality of extracted DNA may pose a problem for further downstream analysis.

While some studies have examined the ability of DNA to be extracted from tissues archives from human medicine (*Ludyga et al., 2012*), few studies have examined the utility of veterinary archives for research. Several factors make veterinary tissue archives useful for pilot and retrospective studies, including natural disease in potential experimental models, fewer regulatory requirements (such as institutional review board approval), and wide availability. However, many veterinary tissue archives have suffered from less than ideal storage conditions and variable tissue processing conditions, which may affect their utility.

While human and veterinary laboratories use the same methods for fixation, there are a number of differences between the two. First, veterinary archives contain a substantially larger proportion of samples from postmortem samples rather than biopsy samples. This may lead to DNA degradation prior to fixation, exacerbating difficulties in isolating DNA from fixed tissues. Even for biopsy samples, many samples are mailed into the laboratory rather than coming from operating suites on site. This may lead to extended fixation times, which also worsen DNA fragmentation. Finally, storage conditions may vary widely in veterinary medicine, with many FFPE blocks being stored, at least for a period of time, in barns or other less controlled environments. These variations in humidity and temperature may further degrade DNA compared to samples in human diagnostic labs. Storage conditions, fixation time, and sample preservation, all play a role in the ability to extract DNA for downstream analysis (*Granato et al., 2014*).

To determine how long tissues sampled and processed routinely in the veterinary diagnostic setting are useful, we isolated DNA from samples in the University of Florida Veterinary Diagnostic Laboratory archive and attempted to generate different amplicon sizes. While less sensitive than real-time PCR, successful agarose gel electrophoreis indicates recovery of DNA amounts sufficient for further downstream processing, such as high-throughput sequencing. This should provide information about the utility of these samples for future research, as well as the best target PCR amplicon size for a given sample age.

**Table 1  DNA primers used in this study.**

| Primer name | Sequence |
|---|---|
| IRBP_F | CCT KGT RCT GGA NAT GGC |
| IRBP_R1_100bp | CTC TTG ATG GCC TGC TC |
| IRBP_R2_200bp | GGC TCA TAG GAG ATG ACC AG |
| IRBP_R3_300bp | CAG GTA GCC CAC RTT NCC CTC |
| IRBP_R4_400bp | CGG AGR TCY AGC ACC AAG G |
| IRBP_R5_500bp | GAT CTC WGT GGT NGT GTT GG |
| SDHD_F | CTACGCGCCCAGATGTTTTC |
| SDHD_R1_100BP | GCACATAAATTGTTATGCCAGTCC |
| SDHD_R2_200Bp | CACATGTGTATGGAACATAGGC |
| SDHD_R3_300Bp | GCCCATTGACCCCGGATT |
| SDHD_R4_400Bp | GAGGCTGTTCCTAAGAGTTG |
| SDHD_R5_500Bp | AGACACACACAGCTCCTTCA |

## MATERIALS AND METHODS

### Tissues

Archival paraffin tissue block were randomly selected from necropsy cases from the University of Florida Veterinary Diagnostic Laboratory using the random number function in Microsoft Excel V14.5.0. To minimize other sources of variation, only cases from dogs were used and the first block containing liver was selected. To try and minimize variations in the time of year, the case range was limited to the first 100 cases per year. All blocks contained tissues fixed in 10% neutral buffered formalin and embedded in paraffin. The tissue archive contains tissues from 1977 until 2015; therefore, five cases were selected for 1977, 1982, 1987, 1992, 1997, 2002, 2007, 2012, and 2015, resulting in a total of 45 cases.

### Extraction of DNA from paraffin blocks

The QIAamp® DNA FFPE Tissue Kit (Qiagen Inc., Valencia, CA, USA) was used for DNA extraction per the manufacturer's protocol. All samples were eluted using 50 μl of buffer ATE and 5 min of room temperature incubation.

### PCR analysis

Each DNA sample was used as a template for PCR amplification, using the primer pairs listed in Table 1 to generate multiple amplicon lengths for the retinol-binding protein 3 (IRBP3) as previously described (*Craft, Conway & Dark, 2014*) as well as for the succinate dehydrogenase subunit D gene. Briefly, samples were amplified using PCR Master Mix (EmeraldAmp® GT PCR Master Mix) on a Veriti Thermocycler (Applied Biosystems, Life Technologies Corp., Burlington, Ontario, Canada) with the following conditions: 96 °C for 3 min, followed by 35 cycles of 96 °C for 1 min, 60 °C for 1 min, then 72 °C for 1 min. This was followed by 7 min at 72 °C, then held at 4 °C Electrophoresis was performed on a 1% agarose gel for visualization of products. DNA previously extracted from tissues frozen at −80 °C using the QIAamp DNA Mini Kit (Qiagen Inc.) and water were used as positive and negative controls, respectively (*Craft, Conway & Dark, 2014*).

## Statistical analysis

Data were analyzed using the Kruskal–Wallis test and the Dunn test with Šidák correction for multiple comparisons in R version 3.2.3 (*R Core Team, 2015*) using the dunn.test package (*Dinno, 2015*). Graphs were generated using the PROC GLM proceedure of SAS (version 9.4; SAS Institute Inc., Cary, NC) and GNUplot v5.0 patchlevel 1.

## RESULTS

DNA was successfully extracted from all samples, as at least one amplicon was generated from each sample. For the IRBP gene, a total of 97.7% (44/45) of samples had a 100bp amplicon generated and 95.5% (43/45) had a 200bp amplicon generated. The success rate dropped with increasing amplicon size—only 35.5% (16/45) of 300bp amplicons, 20% (9/45) of 400bp, and 13.3% (6/45) of 500bp amplicons were generated (Fig. 1A). For the SDHD gene, a total of 95.5% (43/45) of samples had 100bp and 200bp amplicons generated. The success rate decreased with increasing amplicon size; 84.4% (38/45) of 300bp amplicons, 17.7% (8/45) of 400bp, and 26.6% (12/45) of 500bp amplicons were generated (Fig. 1A).

The length of time after processing had a significant effect on DNA amplicon size, as larger samples were generated from tissues from 2015 compared to the remaining samples (Fig. 1B). While four out of five samples from 2015 generated 500bp amplicons, only one additional sample, from 1982, also generated a 500bp amplicon. All but three samples generated 100 and 200bp amplicons from the IRBP gene. The SDHD gene differed from IRBP3 in the 300bp and 500 bp amplicons, although it had a similar trend to IRBP3. When comparing the most recent samples, four out of five amplified the 300bp and 500bp fragment from both SDHD and IRBP3, although the unsuccessful sample was different for each. Only samples from 2015 reliably produced the larger amplicon sizes compared to the other years examined; however, all years except 2002 produced at least one 500bp amplicon (Fig. 1C). Increased age produced shorter maximum amplicon sizes than recent samples. However, other than for samples from 2015, there was not a specific relationship between age and maximum amplicon size (Figs. 2A and 2B). Kruskal–Wallis testing shows a significant difference in maximum amplicon size within the experimental groups ($p = 0.01$). Dunn's test of multiple comparisons with Šidák's correction shows a significant difference between samples from 2015 and 2002, 1997, and 1992 ($p = 0.0185$, 0.0185, and 0.0031, respectively).

## DISCUSSION

FFPE archives are a powerful resource for both retrospective and prospective biomarker discovery (*Magdeldin & Yamamoto, 2012*). Recently, the increasing sophistication of research techniques in examining biological agents and tumor therapy (*Bonin & Stanta, 2013*) have made these more valuable than ever. Polymerase chain reaction (PCR) allows the use of these tissues for downstream analysis into disease pathogenesis. While formalin degradation of DNA can be severe and yielded generally small product sizes in most necropsy samples, more than 95% of samples were able to amplify 200bp long fragments.

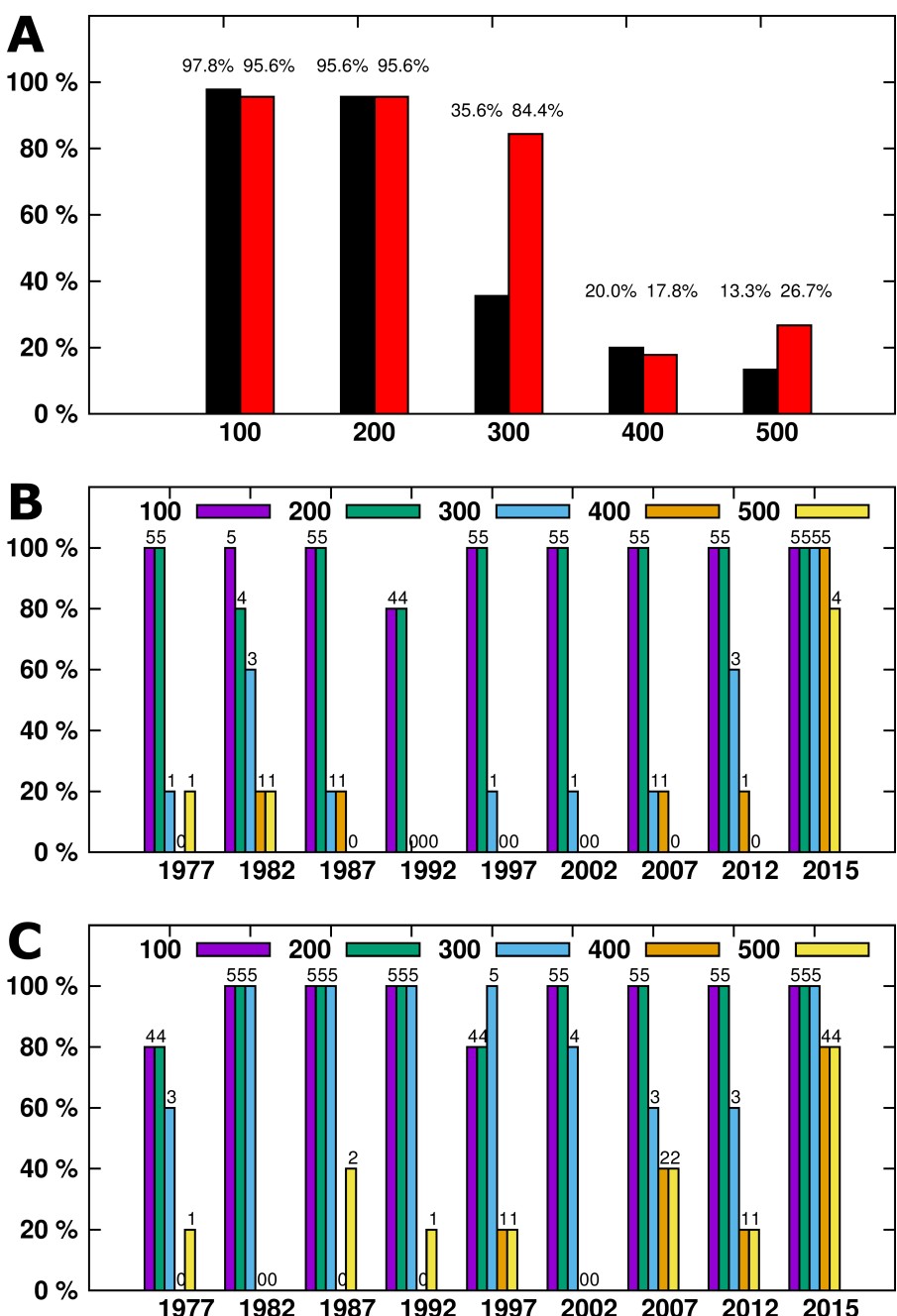

**Figure 1** (A) **The percentage of all samples generating a specific amplicon size. IRBP result arelisted in black, SDHD in red. (B) (IRBP) and (C) (SDHD)- DNA amplicons generated by year.** Numbers above each column indicate the number of samples successfully amplified.

Previous reports in human medicine (*Iwamoto et al., 1996*; *Von Ahlfen et al., 2007*; *Taga et al., 2013*) have found similar results In one study of 25 year-old samples of FFPE tissue specimens between1979 and 1983 processed in 2005, 69% produced 152bp amplicons while 17% and 5% of the samples gave 268bp and 676bp, respectively (*Gillio-Tos et al., 2007*).

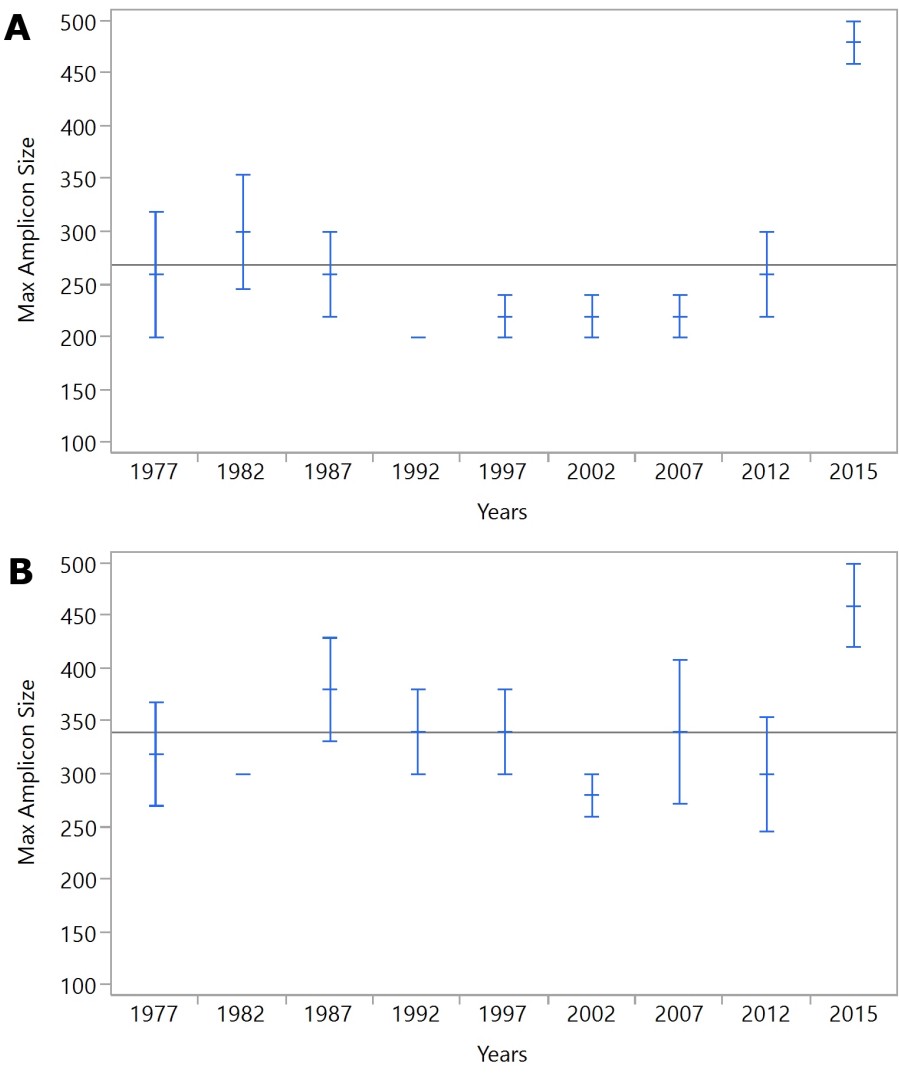

**Figure 2** **(A) IRBP, (B) SDHD.** Mean maximum amplicon sizes for each year are represented by blue horizontal lines. The whiskers indicate the standard error of the mean.

After 50 years, DNA extraction becomes significantly more difficult (*Paireder et al., 2013*). This may be due to a variety of factors, many of which are likely present in our samples. First, samples may have sat in formalin for varying amounts of time prior to processing and paraffin embedding. In veterinary medicine, this is particularly likely with necropsy samples, as the majority of biopsy samples will be sent to the lab immediately and processed in 2–3 days. Second, storage conditions vary, both by year and by sample type. Older samples may be stored in non-climate-controlled conditions, increasing the rate of DNA decay. Finally, samples may have been subjected to a variety of conditions prior to formalin fixation. For necropsy cases, the post-mortem interval and tissue decomposition are significant factors that influence DNA quality. However, despite these challenges, we still obtained useful DNA from more than 95% of the samples examined.

Ideally, tissues placed in formalin for fixation would have fixation times of less than 24 h and would be utilized for short amplicon lengths for optimal molecular analysis (*Turashvili et al., 2012*).

However, the reality of veterinary practice makes this impractical. Samples must be sent from distant practices, be accessioned in the laboratory, trimmed into cassettes, and processed. While alternative fixatives could be useful in increasing the utility of samples for future molecular diagnostics, these have proved impractical as a routine replacement for formalin in the veterinary diagnostic setting (*Craft, Conway & Dark, 2014*). Therefore, this study is important to determine how useful a typical veterinary archive is for retrospective studies. However, this also suggests that veterinary diagnostic laboratories should examine their standard operating procedures regarding tissue processing and formalin fixation to maximize the utility of archives for future diagnostic and research work.

It is unknown if the lack of a specific relationship between age and maximum amplicon size is due to differences in sample quality (such as decomposition), storage conditions, tissue amounts in the blocks, or disease processes. Despite these possible differences, 200bp or smaller amplicons were able to be amplified from nearly all samples. Significantly, this size range is reasonable for many high-throughput sequencing techniques (*Lin et al., 2009*), which makes FFPE samples potentially useful in future genomics studies, especially in analyzing naturally occurring neoplasms.

Although there are differences between the ability to generate amplicons of various sizes between both genes examined, the trend is similar for both genes. However, the SDHD gene was able to amplify 6 additional 500bp samples and 22 additional 300bp samples. This could be due to a number of reasons. First, the IRBP primers contain ambiguity nucleotides, as it was designed to amplify DNA from a number of species, while the SDHD primers were designed specifically for the dog and have no ambiguous nucleotides. Second, formaldehyde reacts with the amino groups of adenine (A), guanine (G), and cytosine (C), forming covalent linkages. Therefore, DNA fragmentation, sequence modification, and cross reactivity are the three major effects of formaldehyde fixation on DNA alteration (*Tang, 2006*). Differences in the amount of A, G, and C in the primers may make IRBP binding sites more susceptible to degradation, making them less likely to amplify. The ambiguity nucleotides may amplify this, as there are fewer specific copies to bind to the smaller number of undegraded template copies.

## CONCLUSION

DNA extracted from archived tissue sample can be useful even in samples over 35 years old. However, amplification of fragments larger than 200–300bp becomes difficult after a few years, although 200bp fragments can be amplified from samples more than 30 years old. Given the large number of samples and occurrence of natural animal models of human disease, veterinary tissues archives should be explored as a source of material for pilot and retrospective studies into the molecular pathogenesis of neoplasms, infectious disease, and genetic abnormalities.

### Funding

The authors received no funding for this work.

### Competing Interests

The authors declare there are no competing interests.

### Author Contributions

- Firas M. Abed conceived and designed the experiments, performed the experiments, analyzed the data, contributed reagents/materials/analysis tools, wrote the paper, prepared figures and/or tables, reviewed drafts of the paper.
- Michael J. Dark conceived and designed the experiments, analyzed the data, contributed reagents/materials/analysis tools, wrote the paper, prepared figures and/or tables, reviewed drafts of the paper.

### Animal Ethics

The following information was supplied relating to ethical approvals (i.e., approving body and any reference numbers):

University of Florida Institutional Animal Care and Use Committee approved IACUC Protocol 201408639.

### Data Availability

The raw data has been supplied as Data S1 and S2.

### Supplemental Information

Supplemental information for this article can be found online at http://dx.doi.org/10.7717/peerj.1996#supplemental-information.

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
