# Peer review of "Determining the utility of veterinary tissue archives for retrospective DNA analysis"

_PeerJ, doi:10.7717/peerj.1996_

## Round 0.1 · original submission · Major Revisions

Please, improve your manuscript according to the reviewer's suggestions.

Reviewer 1 ·

Basic reporting

This is a well-written manuscript investigating the utility of using routinely fixed specimens for retrospective molecular-based investigations. Their investigation confirmed that formalin fixation and time spent in paraffin blocks negatively impacts the quality of DNA within the tissues.

The reference section does not match the requested PeerJ format (e.g., many reference lack a publication year).

Line 36: What fixatives have been tested. A citation is needed here.

Line 143–144: What is the reference to indicate that the 200 bp range is “reasonable for many high throughout sequencing techniques”?

Line 107: It is recommended to change to “significant difference in maximum amplicon within the experimental groups (p=0.01).”

Figure 3: shouldn’t the minimum amplicon size detected be 100 bp for each sample?

Figure 3: Since there can be no value above 500 (y-axis), the y-axis should be limited to 500 (it is recommended to scale the y-axis so the top of the axis is just over 500).

Line 118. There is a missing period.

Experimental design

While the research question is well defined, it fails to fill a well-described knowledge gap. As the authors state, there are numerous reports that have investigated the utility of FFPE tissues as archival material for molecular-based analysis. While the authors state that few reports have investigated veterinary tissue, the authors fail to detail how differences in human and veterinary post-mortem archival steps (e.g., is there a difference in average postmortem interval prior to fixation, are there differences in time of fixation, etc.) may impact the results. Or does this report extend the investigation into even older archival tissue? Without these explanations, there is limited rationale to question the other reports that indicate that archival postmortem material does contain amplifiable DNA, but of a small size. Expansion of the literature review to directly address what is the knowledge gap (e.g., why would veterinary archives be any different) is highly recommended.

Validity of the findings

The experiments are appropriately controlled and the analysis of the data is appropriate.

Additional comments

No comments

Reviewer 2 ·

Basic reporting

The authors investigate the utility of archived formalin fixed paraffin embedded samples for use in retrospective molecular diagnostics studies. They aim to evaluate the effect of length of time since processing on samples. There is sufficient background in the introduction. Three figures are provided. They are all the same data and might better be different parts of the same figure. In addition, it is unclear what is 'predicted' in the third figure. These appear to be averages and no 'prediction' is actually included.

Experimental design

The authors use a single extraction method (QiaAMP DNA FFPE) in order to purify DNA from 5 samples at 5 year intervals. The entire manuscript is based on a single experiment. The methods are significantly lacking in that they do not explain their choice of liver as the tissue of choice or what they are amplifying. They list primers but do not reference the target DNA in the entire article to my knowledge. In addition there is no detail on the PCR conditions. The authors amplify this product and run this on an agarose gel. This seems to be a relatively crude way of detecting amplicon and realtime PCR would likely be more sensitive/quantitative. The selection of only 'IRBP' as an amplicon leaves potential question about whether this is unique to this gene or if a similar phenomenon would be observed in other targets. This could potentially be done on the same DNA extraction and correlation between multiple targets would strengthen the findings.

Validity of the findings

As stated earlier, detection of positive or negative for each of the 5 samples is performed by examining an agarose gel (images not provided) which is less sensitive than realtime PCR. It is possible that at low levels of product, variations in amplicon loaded and researcher evaluation could lead to variation in results. The assumption is that the results for 'IRBP' are generalizable across the genome. Inclusion of another target with high correlation between the two would go a long way in convincing the reader that this is likely. The authors speculate at one point that the issue with older samples is that they may have been in formalin too long before processing. This seems an odd conclusion as it is difficult to explain why tissue handling would have significantly differed 35 years ago.

Additional comments

The evaluation of effect of time of storage on suitability of archival tissue for retrospective molecular diagnostics is useful and should be confirmed relative to previously published literature on human samples. However, more experimentation may be required to do this effectively. Have the authors considered evaluating other tissues as well as other genes? Oncologic studies may involve other tissues and how they compare to liver may be an interesting second question to evaluate and would further increase the value of this manuscript.

---

## Round 0.2 · Minor Revisions

Please consider the remaining points of Reviewer 2.

Reviewer 1 ·

Basic reporting

The authors sufficiently addressed the concerns from the first review.

Experimental design

The authors sufficiently addressed the concerns from the first review.

Validity of the findings

The authors sufficiently addressed the concerns from the first review.

Reviewer 2 ·

Basic reporting

Line 110 has an extra period

Line 157 has an extra space

Line 190 change 'gene' to 'genes'

The new background is substantially better.

Experimental design

The authors did not address the concern that detection of amplification of bands in agarose is likely less sensitive than realtime PCR.

The inclusion of a information about the gene, PCR conditions and a second gene target significantly improve the experimental design section.

Validity of the findings

Figure 2 is still a problem. First, it is unclear what the dots represent. Second, each time point should have five samples and therefore 5 maximum amplicon data points. Whisker plots in this instance seem to suggest more data than is actually present. Would a graph of the mean with standard error suffice? In addition, the line connecting the means conventially suggests repeated measures rather than separate measures. This should be removed.

The conclusion appears to suggest that amplification becomes more difficult over time. This study demonstrates that there isn't increasing difficulty after the initial drop from 2015 to 2012. Perhaps this conclusion could be altered to state that after the initial drop, DNA fragmentation does not significantly alter max amplicon size over the remaining 35 years.

---

## Round 0.3 · accepted · Accept

Congratulations for your manuscript and work.